# Domain Adaptation Methods for Lab-to-Field Human Context Recognition [note 1]

**DOI:** 10.3390/s23063081

**Published:** 2023-03-13

**Authors:** Abdulaziz Alajaji, Walter Gerych, Luke Buquicchio, Kavin Chandrasekaran, Hamid Mansoor, Emmanuel Agu, Elke Rundensteiner

**Affiliations:** 1Data Science, Worcester Polytechnic Institute, Worcester, MA 01609, USA; 2Department of Computer Science, University of Victoria, Victoria, BC V8P 5C2, Canada

**Keywords:** ubiquitous computing, domain adaptation, context aware systems, machine learning

## Abstract

Human context recognition (HCR) using sensor data is a crucial task in Context-Aware (CA) applications in domains such as healthcare and security. Supervised machine learning HCR models are trained using smartphone HCR datasets that are scripted or gathered in-the-wild. Scripted datasets are most accurate because of their consistent visit patterns. Supervised machine learning HCR models perform well on scripted datasets but poorly on realistic data. In-the-wild datasets are more realistic, but cause HCR models to perform worse due to data imbalance, missing or incorrect labels, and a wide variety of phone placements and device types. Lab-to-field approaches learn a robust data representation from a scripted, high-fidelity dataset, which is then used for enhancing performance on a noisy, in-the-wild dataset with similar labels. This research introduces Triplet-based Domain Adaptation for Context REcognition (*Triple-DARE*), a lab-to-field neural network method that combines three unique loss functions to enhance intra-class compactness and inter-class separation within the embedding space of multi-labeled datasets: (1) domain alignment loss in order to learn domain-invariant embeddings; (2) classification loss to preserve task-discriminative features; and (3) joint fusion triplet loss. Rigorous evaluations showed that *Triple-DARE* achieved 6.3% and 4.5% higher F1-score and classification, respectively, than state-of-the-art HCR baselines and outperformed non-adaptive HCR models by 44.6% and 10.7%, respectively.

## 1. Introduction

There is a great potential for context-aware (CA) systems to impact many fields, such as healthcare, smart homes, and security [1]. An important part of CA systems is Human Context Recognition (HCR), the process of determining the user’s current state. Several definitions exist, but ours is as follows: Human Context is a tuple <Activity, Prioception> comprising of the user’s current activity (e.g., walking, standing) and the phone’s placement in the user’s body (the “prioception”) (e.g., in a bag, pocket, or hand). We focus on CA and HCR on smartphones, which are now almost ubiquitously owned and possess a wide variety of sensors such as accelerometers, gyroscopes, and position detectors. There are two popular study designs for collecting HCR datasets for supervised machine learning involving human participants: (1) *scripted* [2] or (2) *in-the-wild* [3]. *Scripted* studies involve participants carrying out a series of tasks in a prescribed sequence while being monitored by a human proctor and having their smartphone sensor data continuously recorded by an app. After the users’ sensor data has been collected, human proctors label the data with the locations they were in. On the other hand, *in-the-wild* studies entail data collection in the subjects’ natural environments over the course of several days. Smartphone sensors collect data in real time, and users are periodically prompted to report their current context so that the app can annotate their collected data. **Motivation.** Due to the high costs associated with gathering labeled data, it is challenging to gather adequate labels of sufficient quality for fully supervised learning in many real-world, in-the-wild scenarios. This presents a challenge for fully supervised learning methods. Thus, it is highly desirable to design innovative learning methods that obviate the need for fully annotated data but instead leverage unlabeled data.

**Problems with In-the-wild datasets that hinder HCR performance:** Due to the high precision of sensor data and high quality context labeling, supervised HCR classification models often achieve excellent precision on scripted datasets. *DeepContext*, a cutting-edge deep learning HCR model, achieved 91.2% accuracy on a scripted dataset [4]. Nevertheless, scripted datasets are not realistic as the contexts visited and visit patterns do not reflect reality. It is important that HCR models are precise on datasets collected in the wild, which are more reflective of actual deployment circumstances. However, HCR models often exhibit reduced performance on more naturalistic, in-the-wild datasets. Vaizman, for example, achieves 71.7% accuracy using a Multi-Layer Perceptron (MLP) model trained on a HCR dataset gathered in the wild [3]. Essentially, the accuracy of state-of-the-art HCR models on scripted datasets decreases by 19.5% when compared to their accuracy on datasets gathered in the wild. This performance disparity is a result of in-the-wild dataset problems, including:

*(1) Diversity of phone placements:* Or locations where smartphone devices are typically kept (prioceptions). When the phone is carried in various prioceptions, sensor signals for the same activity have distinct characteristics [5,6]. In fact, as seen in Figure 1, prioception is one of the most significant causes of variation in smartphone context sensor data [3]. Smartphone users may opt to carry their devices in a backpack, in their hand, or in a coat pocket when engaging in a certain activity (e.g., walking).

*(2) Weak, noisy, and missing context labels:* Supervised machine learning algorithms have difficulty when users quit supplying labels [7] due to lack of time or because they offer incorrect labels [8].

*(3) Diversity of smartphone models:* Typically, while participants in scripted HCR studies use a specific study phone model provided by the proctor, participants in HCR field (in-the-wild) studies utilize their own mobile devices. Different smartphone models can record sensor values for the same context that vary by as much as 30%, which presents an additional challenge for machine learning classifiers [6].

**Lab-to-field methods:** Have recently been proposed as promising approaches for achieving acceptable HCR performance on noisy, low-quality labeled datasets collected in the wild [9]. The goal of lab-to-field techniques is to train highly accurate machine learning models on scripted HCR datasets, which are subsequently adapted for *in-the-wild* datasets with the expectation of maintaining great performance. *(Going from Lab-to-field)*. Due to differences between the contexts visited by participants in the scripted study visited vs. the in-the-wild study, as well as the order in which they visited those contexts and the length of their visits, which were quite different from in-the-wild scenarios, there are substantial discrepancies between the distribution of features derived from scripted and unscripted datasets, often known as the *covariate shift problem* [9,10,11].

**Domain Adaptation (DA):** One of the primary approaches of adapting neural networks to deal with the covariate shift problem is called Domain Adaptation (DA), and it is a transductive transfer learning method. DA has been utilized in other related disciplines, such as object detection in computer vision and the issues posed by the unpredictability in wearable sensor location in ubiquitous computing [5,11]. Utilizing both labeled source samples (e.g., scripted dataset) and unlabeled target samples (e.g., in-the-wild), each of which has a unique distribution, Unsupervised Domain Adaptation (UDA) attempts to learn a deep learning model that can accurately predict the labels of unlabeled (e.g., in-the-wild) data samples [5,12]. Figure 2 presents an overview of the topic, its obstacles, and our strategy.

**Challenges.** For the application of UDA to the lab-to-field generalization of smartphone context recognition, two significant obstacles must be overcome. Initially, the previously noted data concerns with in-the-wild datasets (the diversity of phone placements, noisy labels, and the variety of smartphone models) must be resolved. Second, it is difficult to build a strategy for knowledge transfer from a scripted dataset to a more realistic, but significantly noisier, unscripted dataset with sparse labels.

**Our approach.** Recent empirical accomplishments of the triplet loss function in the facial recognition task [13,14], where changes of the same person’s face pictures are tightly mapped in a learned embedding space, has inspired us. We believe that HCR sensor data can benefit from a similar approach even in cases where sensor signatures associated with the same context often vary. Our opinion is also consistent with the findings of Khaertdinov et al., who utilized triplet loss to reduce the impacts of subject variability and enhance model generalizability [15].

We present *Triple-DARE*, a Lab-to-field UDA approach that may harness the vast volumes of unlabeled smartphone HCR data collected in the wild, therefore reducing the requirement for human-annotated labels. *Triple-DARE* uses both handcrafted features and features autonomously extracted from raw sensor data by a CNN. *Triple-DARE* uses domain alignment and triplet losses to learn domain-invariant embeddings with discriminative capabilities for context predictions learned from unlabeled samples. *Triple-DARE* collects domain-invariant features that increase the effectiveness of predicting contexts under unobserved prioceptions.

In addition, to support our DA strategy, we used HCR datasets with coincident scripted and in-the-wild data with equivalent context labels collected in both studies [1]. These coincident datasets and identical context labels guarantee that there is a feature representation of contexts that is shared throughout scripted and unscripted datasets, which is a crucial need for the DA strategy. By only using context labels that were collected in a scripted study during model development, we are able to demonstrate that our method is applicable to HCR models that are implemented in realistic environments by employingDA in order to reduce the impact of potentially noisy labels while maintaining HCR performance on a dataset collected in the wild. *Triple-DARE* outperforms state-of-the-art baselines by 3.79% and 1.89% gains in F1-score and accuracy, respectively, and outperforms HCR models without *Triple-DARE* by 39% and 14.7% in F1-score and accuracy, respectively.

**State-of-the-art limitations.** There is a paucity of research on laboratory-to-field generalization approaches for HCR. Previously proposed lab-to-field methods include importance re-weighting [9,16] and Positive Unlabeled (PU) classifiers [1]. DA has been used in the past to solve the problem of variable on-body locations of wearable sensors [5,17] but not for HCR. The majority of prior DA work for wearable sensors focuses on decreasing the global distribution gap across domains while learning common feature representations [5,17]. However, we observe that even if the global distribution is effectively aligned, samples from different domains with the same label may be mapped such that they are far apart in feature space. Thus, in addition to using a domain alignment loss [18,19], *Triple-DARE* improves intra-class compactness and inter-class separability by utilizing a joint fusion triplet loss [12,13] intended for multi-labeled datasets. Moreover, unlike other existing methods for dealing with domain shifts [1,9,17,20], we do not utilize target labels in the target (in-the-wild) dataset, instead following the UDA problem setting outlined by Chang et al. [5].

**Contributions.** The main contributions of this paper are:1.We provide *Triple-DARE*, a novel UDA deep-learning architecture that employs a scripted dataset to increase the HCR accuracy of predicting contexts in the wild. *Triple-DARE* employs a domain alignment loss for domain-independent feature learning, a classification loss to keep task-discriminative features, and a joint fusion triplet loss to improve intra-class compactness and inter-class separation;2.We carefully assessed *Triple-DARE*, comparing it to numerous state-of-the-art unsupervised domain approaches, including DAN [18], CORAL [19], and HDCNN [17], and bench-marking advances in HCR performance on target domains in multiple application scenarios. Our ablation study demonstrates that all component of *Triple-DARE* contributes non-trivially;3.We illustrate that *Triple-DARE* minimizes in-the-wild dataset problems when compared to state-of-the-art DA algorithms, delivering improved prediction accuracy on the target (in-the-wild) domain without the requirement for large amounts of source-labeled samples.

The rest of this paper is organized as follows. Section 2 includes the background. Section 3 reviews the related work. Section 4 describes our proposed approach. Section 5 presents our evaluation and findings. Section 6 outlines the limitations of our work and plans for future work. Section 7 finally concludes the paper.

## 2. Background

### 2.1. Covariate Shifts

The term “Covariate Shifts” was first introduced by Shimodaira [21], and is described as changes in the distribution of the input *x*. While there are other types of existing shifts [10], the most researched type of shift is covariate shift. Covariate shift occurs when data are generated based on a model P(y|x)P(x) whose distribution P(x) varies between the training and test situations. While there is some ambiguity in the definitions of covariate shift in the literature, we found the definition provided by Moreno-Torres et al. [10] to be the most relevant, given by the following conditions:(1)$$P_{\mathrm{training}}\left(y|x\right)=P_{\mathrm{testing}}\left(y|x\right)\;\mathrm{and}\;P_{\mathrm{training}}\left(x\right)\neq P_{\mathrm{testing}}\left(x\right),$$
where $P_{\mathrm{training}}\left(x\right)$ and $P_{\mathrm{testing}}\left(x\right)$ represents training and testing input distributions, respectively.

Collecting smartphone sensor data in the wild often results in naturally occurring variations in the data. When trying to leverage models trained on scripted data to improve performance on an in-the-wild dataset with similar context labels, we encounter a data shift problem known as covariate shift, where the distribution of features differs across training and test scenarios. Specifically, the covariate shift problem is caused by substantial differences between the distributions of features extracted from scripted and in-the-wild datasets [9,10,11]. More broadly, because real-world applications must face some type of dataset shits, it is critical to address the covariate shift problem for the successful deployment of machine learning models in the wild [10].

### 2.2. Sensor Data Collection Studies

Inaccurate labeling or unrealistic user behavior are two common problems with context datasets. There are two types of research designs used to gather HCR datasets: *scripted* [2] or *in-the-wild* [3]. *Scripted* studies are usually conducted in a laboratory setting. Participants follow a scripted series of steps to complete a series of tasks in a predetermined order, while an app on their smartphones continuously logs data from those devices’ sensors. Human proctors annotate the sensor data with corresponding context labels. In *unscripted* (“or in the wild”) studies, data is gathered over days while people live their lives in the actual world. A smartphone continually gathers smartphone sensor data as individuals go about their daily lives. Periodically, subjects annotate their data with labels for the contexts they have visited. While the scripted technique for HCR data collection produces accurate labels suited for supervised machine or deep learning that are exceptionally precise and consistent, the contexts visited and sensor data acquired in each context are not reflective of the actual world. HCR research conducted in the wild yields more accurate results. However, certain context labels may be missing since people forget to label when their lives get busy. Additionally, some labels may be incorrect due to human labeling errors [8].

### 2.3. DARPA WASH Project: Motivation Use Case

The Warfighter Analytics utilizing Smartphone for Healthcare (WASH), a DARPA-funded project, investigates passive smartphone evaluation of traumatic brain injury and infectious disease [22]. This will offer a current evaluation of the warfighter’s battle readiness. Initially, the target groups are active-duty military personnel and veterans, but the findings will also apply to civilians. In the intended use case, the WASH smartphone application will passively collect sensor data throughout each day. Each day’s data is then sent overnight to the cloud for processing. These data will be analyzed in the cloud by disease inference algorithms to provide a *bioscore* (or probability of illness) for each warfighter.

**Program phases:** The WASH program is separated into two sections. Phase one is identifying particular smartphone user scenarios for conducting targeted health evaluations. Phase two entails the development of real TBI and infectious illness assessment systems for smartphone users. In phase one, we conducted research and compiled a list of smartphone biomarkers indicative of TBI and infectious disorders, as well as their accompanying settings. Our team performed user surveys to acquire labeled data for these settings and developed HCR models to infer these smartphone contexts derived from labeled sensor data. In Table 1, the intended disease-specific tests or biomarkers relating to each of these settings are detailed. The University of Massachusetts Medical School’s specialists in traumatic brain injury (TBI) and infectious diseases were consulted while compiling our list of illness tests and situations (UMMS). As an example, trembling hands are a symptom of TBI. In  phase one, our team will perform user research and develop deep learning models to recognize smartphone users holding their devices. In the second step, we will analyze if the user’s hand is shaking. This study is limited to context recognition. Actual context-specific disease assessment research is not covered.

### 2.4. Our Coincident Data Gathering Study Approach

Using an innovative coincident study design, we conducted scripted and in-the-wild data collection studies to collect labeled data in the same contexts shown in Table 2. This coincidental study enables the use of machine learning techniques that combine the precision of scripted labels with the natural context visit patterns of studies conducted in the wild. Our in the wild study followed a similar methodology to the *Extrasensory* study. The smartphone app constantly collected sensor data from 103 participants’ smartphones as they went about their daily lives. The users were subsequently prompted to self-report context labels. Our scripted study was conducted in a specific laboratory, campus building, or route. The smartphone app systematically collected data from 100 participants who visited the contexts listed in Table 2. The scripted data collection session lasted approximately one hour per subject, and human proctors monitored and annotated the data manually.

### 2.5. Weakly Supervised Learning (WSL)

In supervised learning tasks, predictive models are trained on annotated training examples, common types of which are classification and regression models. A training example consists of an input feature vector (also known as an instance) and a label that is associated with it (or ground-truth). Due to the high costs associated with gathering labeled data, it is difficult to gather adequate labels of sufficient quality for fully supervised learning in many real-world scenarios, such as our study of HCR using data collected in the wild. This presents a challenge for fully supervised learning. Various types of weak (or inaccurate) labels can occur in such practical scenarios, including several encountered in our mobile HCR scenarios, requiring innovative learning methods. According to a recent survey by Zhou et al. [7], weakly supervised learning can be categorized into three types:1.*Inexact supervision* in which only coarse-grained labels are provided. Due to the nature of the annotation process of sensor data, only a few selected sub-segments of each training sensor segment can be considered accurate representatives of their respective labels. However, their precise length, as well as their position within the segment, is unknown;2.*Inaccurate supervision* in which data labels are not always correct. For example, in-the-wild datasets often depend on self-reported labels. However, users may erroneously provide wrong labels as they might not recall which contexts they previously visited accurately;3.*Incomplete supervision* that utilizes unlabeled training data. When study participants get busy with their lives, they might forget to label the data in the dataset, which means that some of the context labels might be missing from the dataset.

For these various forms of weak labeling, innovative learning methods that are trained under weak supervision are desired [7].

## 3. Related Work

**Lab-to-field generalization.** Our Lab-to-field method tries to leverage a scripted dataset that contains high-quality, relatively cheaper to obtain, ground truth labels to improve HCR model performance on an in-the-wild dataset [9]. The ability of deep neural networks to generalize to real-world scenarios, where domain shift is expected, is a critical challenge in smartphone HCR developed for in-the-wild data [1,23]. Importance re-weighting [9,16] and Positive Unlabeled (PU) classifiers [1] are two methods that have been presented in the past to deal with covariate shifts. The transferability of HCR findings from the laboratory to the real world is an area that has received little attention. However, a related study employed importance re-weighting to modify a linear logistic regression model for application with data from wearable electrocardiograms (ECGs). When applied to deep neural networks, however, these techniques have a diminished impact on performance [24]. Unlike other existing methods for dealing with domain shifts [1,9,17,20], our approach does not require target domain labels.

**Domain Adaptation (DA).** Prior research has demonstrated substantial progress in adapting deep neural networks to various related domains [11]. Recent deep DA methods are either discrepancy-based approaches that minimize a discrepancy metric over feature distributions [18,19], or adversarial-based approaches [25] that aim to maximize domain confusion. The Deep Adaptation Network (DAN) [18] minimized the mean distance between two feature distributions in a Reproducing Kernel Hilbert Space (RKHS), effectively matching higher-order statistics of the two distributions. On the other hand, the deep Correlation Alignment (CORAL) [19] technique proposed matching the mean and covariance of two distributions. Other strategies have used an adversarial loss to maximize domain confusion [25]. The domain alignment loss, one component we utilized in *Triple-DARE*, is based on DAN.

**DA for wearable sensor data.** In ubiquitous computing, several DA techniques have been developed to transfer a trained model to a new dataset with similar characteristics [5,17,26,27]. Previous work has shown that DA can be used to unsupervisedly learn domain-invariant accelerometer [5,17] and gyroscope [5] features from sensor data by minimizing a discrepancy distance in the Convolutional Neural Network (CNN) embedding, thereby mitigating the effects of variability in wearable sensor placement. HDCNN [17] looked at whether or not a model pre-trained on smartphone data could be used with unlabeled smartwatch data. The researchers used Kullback–Leibler (KL) divergence and a discrepancy-based technique to transfer the trained model from smartphones to the unlabeled wristwatch data. Stratified Transfer Learning (STL) [26] is a DA method for adapting on-body sensor-based activity recognition tasks to various sensor placements (wrist, chest, leg, etc.). It also maps source and target domain data into the same subspace where distances can be computed, exploiting intra-affinity of classes to transform intra-class knowledge. UDA methods based on Variational Auto Encoders have been used for adapting models to work on another dataset, and have been applied on binary sensors for smart-homes applications [27]. DA was also used to adapt models to subject variability [20], using multi-domain adaptation to address target label shift by incorporating the target domain label distribution in the training process.

The majority of existing work solely focuses on domain-general feature representation learning with the goal of decreasing the global distribution disparity [5,17]. While STL proposed a way to perform intra-class transfer by minimizing the discrepancy between feature distributions of instances of the same class, this approach does not scale well to large-scale datasets, especially datasets with a large number of class labels. By employing a joint fusion triplet loss, our study expands upon previous efforts to enhance intra-class compactness and inter-class separability [12,13]. A summary of the related work is included in Table 3.

## 4. Proposed *Triple-DARE* Methodology

### 4.1. Problem Formulation

Our study makes use of two datasets annotated with the same context labels as those in Table 2: (1) a scripted dataset (source) with high-fidelity labels, and (2) an in-the-wild dataset (target) with identically labeled data. These two datasets are acquired using a study design with coincident data collection, where data was gathered for the same context labels in separate scripted and in-the-wild settings. With respect to UDA, there are source domain labeled samples and target domain unlabeled samples, both of which have distinct data distributions. Our objective is to use both labeled and unlabeled target data to train a classifier that performs effectively on the target domain. In a more formal sense, we have labeled samples $D_{s}={\left\{\left(x_{i}^{s},y_{i}^{s}\right)\right\}}_{i=1}^{n_{s}}$ and unlabeled samples $D_{t}={\left\{\left(x_{i}^{t}\right)\right\}}_{i=1}^{n_{t}}$, with  $n_{s}$ and $n_{t}$ standing for the number of samples in the source and target domains, respectively. The feature space and label space are identical across the source and target domains $X_{s}=X_{t}$ and $Y_{s}=Y_{t}$, but the marginal distribution is not the same ($P_{s}\left(x_{s}\right)\neq P_{t}\left(x_{t}\right))$. The conditional distributions are assumed to be equal $P_{s}\left(y_{t}|x_{s}\right)=P_{t}\left(y_{t}|x_{t}\right)$ in the two domains.

We refer to x as the feature vector, and y as a multi-label output vector representing the human context where each label used is a binary output (e.g., walking vs. not walking). Presumably, the source and target tasks are identical. At first, we use the labeled source dataset to train the HCR model. Once the HCR model has been trained, it may be used to detect unlabeled contexts in the target dataset by integrating unlabeled data from the target dataset into the training set.

### 4.2. Overview

In the model in Figure 3, the framework of *Triple-DARE* is depicted. *Triple-DARE* extracts two distinct kinds of feature sources from the source and target datasets: (1) Hand-crafted features based on temporal and spectral information processed by a feed-forward neural network and (2) three-axis sensors’ raw data is put into a convolutional neural network (CNN) that uses a soft attention method to identify prominent characteristics from the data. *Triple-DARE* consists of three main parts: (1) A domain alignment loss $L_{d}$ to generate domain-invariant embeddings; (2) a classification loss $L_{\mathrm{cls}}$ in order to preserve task-discriminating characteristics; and (3) a joint fusion triplet loss $mathcalL_{\mathrm{tri}}$ that improves intra-class compactness and inter-class separation in the learned embedding space by learning comparable contexts represented by differences in sensor inputs.

The final result is used for context predictions with multiple labels. For instance, according to our definition of context as an <Activity, Phone placement>, a context could be 〈“Sitting”, “In Bathroom” with “Phone In Hand”〉. Our ultimate aim is to minimize the cost function C(·) in order to execute context predictions by learning discriminative and domain-invariant embeddings:(2)$$\begin{array}{c} C\left(\theta \right)=\lambda_{1}L_{\mathrm{cls}}^{\theta }+\lambda_{2}L_{d}^{\theta }+\lambda_{3}L_{\mathrm{tri}}^{\theta }, \end{array}$$
where θ are model parameters, $\lambda_{1}$, $\lambda_{2}$, and $\lambda_{3}$ are balancing coefficients. Subsequent sections elaborate on this procedure and each kind of loss objective. Each loss function is applied to the three feature encodings produced by our deep network, namely the MLP, CNN, and joint fusion encodings.

### 4.3. Feature Generation

From the raw sensory inputs for a specific smartphone context dataset, we generate two views. The first is a vector derived by manually applying handcrafted features to all accessible sensors. The second view is comprised only of raw three-axial sensors. We use distinct feature encoders for each input view type. Specifically, (1) Handcrafted feature encoding using a Multi-Layer Perceptron (MLP), which is adopted from Ref. [28] and (2) attention-based CNN encoder [4] for raw sensor data. The two resulting feature encodings are then concatenated to yield a joint fusion encoding.

We use data from five sensors: accelerometer, gyroscope, GPS, magnetometer, and phone status (discrete attributes such as whether the phone screen is locked or unlocked). At the sliding window level, we compute statistical, time-based, and frequency-based features for each of the sensor modalities (10-s in this application). Then, we implemented Z-score normalization $z_{i}=\frac{x_{i}-\bar{x}}{s}$ through subtraction of the mean and division by the standard deviation. Handcrafted features, including 188 features borrowed from Ref. [3], are utilized to build a vector that is then put into a feed-forward network. Table 4 lists some of the handcrafted features incorporated in our work.

CNN’s auto-learning capabilities employ raw sensor data from three axial sensors (accelerometer, gyroscope, and magnetometer). The CNN we utilized, which was adapted from the DeepContext [4], has a soft attention mechanism that aids in the learning of prominent features by assigning greater priority to those parts of the raw sensor data that are more indicative of the user’s context. The intuition behind the design of this attention mechanism is similar to that proposed by Refs. [4,30]. The effectiveness of this architecture comes from using attention layers on features generated by single-sensor CNNs and features generated by CNNs that assessed the combined sensor outputs. This enables the model to emphasize CNN characteristics that are context-specific. For more details about the DeepContext CNN architecture, we refer the reader to Ref. [4].

### 4.4. Domain Alignment Loss

The objective of domain alignment loss is to transform the source and target feature encoding into a common feature distribution space in order to discover feature representations that are shared across domains. Gretton et al. [31] presented Multi Kernel Maximum Discrepancy Mean (MK-MMD) as an improvement for Maximum Discrepancy Mean (MMD), which we employ in our method. MMD is a non-parametric distance metric that can be employed to evaluate the disparity between marginal distributions [18]. MMD maps the feature representations of the source and target domains ($X_{s}$ and $X_{t}$) to the Reproducing kernel Hilbert space (RKHS) and then computes the mean distance between the two distributions in RKHS. MK-MMD has been proposed as an optimal kernel selection approach for MMD because it can find an ideal kernel created by a weighted combination of various kernels based on the source and target datasets [18]. Let ϕ(·) be a feature map defined as a combination of G positive kernels $k_{u}$ with their associated bandwidth $\beta_{u}⩾0$, given as the following:(3)$$k=\sum_{u=1}^{G}\beta_{u}k_{u},$$
(4)$$\phi \left(x^{s},x^{t}\right)=k\left(x^{s},x^{t}\right),$$
where $x^{s}$ and $x^{t}$ represent feature embeddings for the source and target domain, respectively. The formulation of MK-MMD is thus defined as follows:(5)$$\begin{array}{c} q\left(X^{s},X^{t}\right)={∥E_{X^{s}}\left[\phi \left(x^{s}\right)\right]-E_{X^{t}}\left[\phi \left(x^{t}\right)\right]∥}_{H}, \end{array}$$
where ${∥.∥}_{H_{k}}$ is the RKHS norm. The domain alignment loss can be obtained by:(6)$$\begin{array}{c} L_{d}^{\theta }=\sum_{l\in N^{l}}q^{2}\left(X_{l}^{s},X_{l}^{t}\right). \end{array}$$
MK-MMD is computed per network layer to measure the distance between the source and target domain data representations. $N^{l}$ indicates the number of layers, and we denote $\left(X_{s}^{l},X_{t}^{l}\right)$ for the distributions of the source and target domains, retrieved from the *l*th layer in the network. $d\left(X_{s}^{l},X_{t}^{l}\right)$ is the MK-MMD calculated by Equation (Equation 5) between the source and target domains distributions evaluated on the *l*th layer embeddings. Intuitively, the domain alignment loss is a regularizer that minimizes the distance between the distributions generating source domain data and target domain data.

### 4.5. Classification Loss

The objective of classification loss is to use source domain labels to discover discriminative features for context predictions. Both domains utilize the same context labels for classification. The overall learning process is guided by the optimization of our model for context classification on the source domain. Given the availability of $D_{s}$’s labels (labeled source domain data), the classification loss is defined as:(7)$$\begin{array}{c} L_{\mathrm{cls}}^{\theta }=\frac{1}{N_{s}}\sum_{i=1}^{N_{s}}\ell_{\Psi }\left(f_{\phi }\left(x_{i}^{s}\right),y_{i}^{s}\right), \end{array}$$
where the classifier is denoted as $f_{\phi }\left(\cdot \right)$, $N_{s}$ represents the number of labeled training samples, $\ell_{\Psi }$ is a binary cross-entropy function with inverse class frequency weighting that corrects for class imbalance, and $\left(xi^{s}\right),y^{s})$ represents labeled context data sampled from source domain data.

### 4.6. Triplet Loss

In an embedding space, the triplet loss is primarily utilized to group samples from the same or related classes together and push samples associated to different classes away. An empirical success was seen in the field of face recognition. This is because different images of the same person map very closely to the learned embedding space [13,14]. Since numerous variations in the sensor inputs can represent the same context, we think the same approach can be applied to sensor data.

Given three samples, an anchor sample $x_{a}$ (also called a query sample), a positive sample $x_{p}$ (one that belongs to the same class as the anchor), and a negative sample $x_{n}$ (i.e., a sample with a different class from the anchor), and with a distance function *d*, we can define triplet loss as follows:(8)$$\begin{array}{c} L_{\mathrm{tri}}^{\theta }=\sum_{i}^{N}{\left[d\left(x_{a}^{i},x_{p}^{i}\right)-d\left(x_{a}^{i},x_{n}^{i}\right)+\alpha \right]}_{+} \end{array}$$
where α represents the margin between positive and negative samples and *x* represents an embedding of *x* for ease of notation. We reduce the triplet loss by pushing $d(x_{i}^{a},x_{i}^{p})$ towards zero and making $d(x_{i}^{a},x_{i}^{n})$ to be greater than $d(x_{i}^{a},x_{i}^{p})+\alpha$. In other words, pairs of positive samples are jointly grouped together, while positive and negative sample pairs are separated by some margin α. To put this in perspective, we want the network to learn a feature space in which the squared distance between all feature embeddings of the same context is minimal, while the squared distance across sensor contexts associated with different labels is large.

### 4.7. Joint-Fusion Triplet Mining

The process of constructing triplets (anchor, positive, and negative) for triplet loss calculations is known as triplet mining. The two main strategies for selecting triplets are offline and online. Finding triplets offline is not recommended as it requires a complete pass over the training set [14]. In accordance with the method described in Ref. [13], we employ an online triplet mining strategy that does not require a prior pass on the training set. Because discovering triplets across two domains necessitates the presence of target domain labels, one of the most prevalent solutions for UDA problems is to use the classifier trained on the source domain to generate pseudo labels for samples of the target domain during training [12]. During this procedure, it is vital to remember that the pseudo labels generated may not be accurate. Nonetheless, we reassign pseudo labels every few iterations since the accuracy of the classifier on the target dataset improves continuously throughout training. In addition, domain alignment loss can help improve the accuracy of the classifier on the target dataset by reducing distribution disparity. Consequently, the quality of the pseudo label can improve automatically.

Our joint-fusion triplet mining technique operates as follows: After concatenating two mini-batches of samples from the source and target domains into one mini-batch, triplets are generated. We need a concept of similarity between multi-labeled vectors in order to construct triplets that are compatible with our multi-labeling settings. First, we define a compatibility score between two binary labeled contexts $y_{1},y_{2}$ as the dot product between them:(9)$$\begin{array}{c} c\left(y_{1},y_{2}\right)=y_{1}\cdot y_{2} \end{array}$$

Due to the imbalanced nature of our dataset, we consider all the positive examples when constructing triplets. We use a strategy similar to Ref. [13] that focuses on triplets that contribute the most to the learning process, but we modify it by using our compatibility score to select triplets that meet the following condition: (10)$$\begin{array}{c} d\left(x_{a},x_{p}\right)+\alpha >d(x_{a},x_{n})\;\&\;c\left(y_{a},y_{p}\right)>c\left(y_{a},y_{n}\right) \end{array}$$

Our triplet mining strategy is detailed in Algorithm 1.
**Algorithm 1:**Joint-fusion online triplet mining finds triplets with multi-labeled vectors**Input**: Number of samples in a batch *m*, classifier f(·), $D_{s}$, $D_{t}$, distance *d*,   compatibility score (Equation (Equation 2)) *c*, α, and sample size *k***Output**: List of triplets [(a, p, n)]${\left\{x_{i}^{s},y_{i}^{s}\right\}}_{i=1}^{m}$← Read next mini-batch ($D_{s}$);${\left\{x_{i}^{t}\right\}}_{i=1}^{m}$← Random sample mini-batch ($D_{t}$);${\left\{y_{i}^{t}\right\}}_{i=1}^{m}$←Assign pseudo labels using *f* on ${\left\{x_{i}^{t}\right\}}_{i=1}^{m}$;${\left\{x_{i}^{z},y_{i}^{z}\right\}}_{i=1}^{m}$← concatenate( ${\left\{x_{i}^{s},y_{i}^{s}\right\}}_{i=1}^{m}$, ${\left\{x_{i}^{t},y_{i}^{t}\right\}}_{i=1}^{m}$);triplets← {}
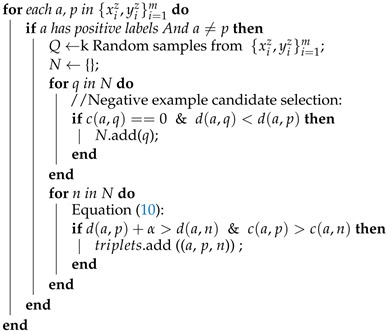
**return** triplets

## 5. Experiments

We compared Triple-DARE and baseline models on both scripted and in-the-wild smartphone HCR datasets, where we performed multiple UDA use cases. Overall, *Triple-DARE* was used to obtain a robust representation from the scripted dataset (source), which was then applied to enhance HCR on the in-the-wild dataset (target).

### 5.1. Datasets

*In-the-wild dataset:* A total of 103 participants downloaded a smartphone app that passively collected data for 2 weeks as they went about their daily lives. Periodically, participants were asked to self-report the context labels that they visited. In addition to being more realistic, our in-the-wild dataset reflected a variety of manufacturer hardware because it was collected using individuals’ smartphones.

*Scripted dataset:* The smartphone application collected information from 100 participants who visited predetermined locations. During the data collection session, which lasted about an hour per subject, human proctors oversaw and manually annotated the data. Both scripted and unstructured datasets were similarly preprocessed and characterized. The contexts were treated as vectors with multiple labels, with 10-s window size used to create segments. The number of samples for scripted and unscripted datasets is 21,846 and 631,026, respectively. In Table 5, we list the context labels used throughout this study. To increase the applicability of our model to unseen subjects, subject-wise cross-validation was used, wherein a given subject’s data was included in either the training set or the test set, but never both. Each UDA experiment utilized 90% of source domain data for training, 10% of source domain data for validation, and 100% of target domain data for testing. Figure 4 displays data extracted from the two datasets, displaying only the accelerometer sensor readings for three context examples.

### 5.2. Baselines

We compared *Triple-DARE* with cutting-edge deep-learning-based DA models: (1) *CORAL* [19]: A cutting-edge UDA model that uses deep-coral discrepancy loss. (2) *DAN* [18]: A model with just our MK-MMD alignment loss. (3) *HDCNN* [17]: An advanced baseline DA technique was previously applied to smartphone sensor data. *HDCNN* cross-domain transfer learning model that uses KL divergence loss on the acquired feature vectors. (4) *SOURCE*: A model that has only been trained in the source domain, with no adjustments made for the target domain. Our proposed model, *Triple-DARE*, which employs our joint-fusion triplet loss.

### 5.3. Implementation and Experimental Settings

*(1) Hyper-parameters:* Grid search was used to optimize the hyper-parameters of MLP and CNN. The learning rate is initialized at 1 × 10${}^{-1}$, balancing coefficients are initialized as $\lambda_{1}=1$, $\lambda_{2}=0$, and $\lambda_{3}=0$. Following the schedule outlined in Ref. [25], the balance coefficients and the learning rate are raised or lowered, making our model more confident in source labels and less sensitive to low-quality pseudo labels during the early phases of training.

The batch size is set to 256. Adam optimizer was used. All trials use the same backbone layers utilized by our DA technique: Two-layer handcrafted-features MLP with 16 hidden dimensions, single-layer MLP domain classifier with 32 hidden dimensions, and a convolutional neural network with attention blocks for individual and combined sensor layers, then an average pooling layer, adopted from Ref. [4]. All sensor data was fed into a three-layer CNN. The outputs are then concatenated and sent to another 3-layer CNN. Attention blocks are utilized to concentrate on prominent regions of inputs [4,30]. In triplet mining, pairwise distances are computed using Euclidean distance and α is set to 0.1. Following the LeakyReLU activation in the final context prediction layer is the Sigmoid activation.

*(2) Evaluation Protocol:* In addition to reporting classification accuracy, we used the *F1* metric to evaluate HCR performance in the UDA setting due to the class imbalance in our context datasets. As the sizes of the source and target domain datasets may not be identical, random sampling is used to iterate through the target domain dataset. However, we evaluate our model on every sample in the dataset for the target domain.

### 5.4. Results and Findings

*(1) Notations*: First, we define the notations used in our experimental results. $S_{\mathrm{Prioception}}$ is denoted for the scripted context dataset and $W_{\mathrm{Prioception}}$ for in-the-wild dataset, e.g., $S_{\mathrm{Bag}}$ refers to scripted contexts, annotated with “Phone In Bag”.

*(2) Overall Results*: In Table 6, we compare the overall performance scores of our *Triple-DARE* algorithm to those of the baseline models. In the overall UDA tasks and Lab-to-field UDA tasks, *Triple-DARE* outperforms the baseline methods with a 4.5% increase in F1-score and 6.3% increase in classification accuracy. Figure 5 displays the performance per context label across all UDA tasks, demonstrating that our approach outperforms state-of-the-art methods across multiple context labels. In general, UDA methods have an advantage over classifiers that are trained solely on the source domain without leveraging unlabeled data. Particularly, UDA methods were of great assistance with the Jogging, Running, Going Up and Down Stairs labels, for which the user is unlikely to provide labels while performing these activities in the wild. Nevertheless, our method makes use of the high-fidelity labels acquired during the scripted study and enhances adaptation. As shown in Table 5, predictions for the labels Sitting and Walking are the most difficult, which may be due to a significant difference in target label distributions.

*(3) Scripted contexts with cross-prioception UDA tasks*: Figure 6 demonstrates that *Triple-DARE* consistently outperforms the baseline methods on all cross-prioception UDA tasks. The adaptation procedure has significantly benefited the UDA tasks with “Phone In Hand” as their target domain. This advantage is a result of the signal noise introduced when the phone is in motion. In the majority of instances, *CORAL* performs better than *DAN*.

*(4) Lab-to-field generalization UDA tasks*:

Figure 7 displays the results of our lab-to-field UDA generalization tasks. Massive differences in the scores obtained for “Phone in Pocket” versus “Phone in Bag” and “Hand” provide additional information about diversity placements. We hypothesize that when the phone is placed in a bag or held in the hand, the model is unable to map data from scripted and in-the-wild datasets to a common feature space. In adapting models learned on scripted data to make context predictions on in-the-wild data with a “Phone in Pocket” prioception, however, we observe a notable improvement over baseline methods of the current state of the art.

*(5) Training under insufficient labels*:

As shown in Table 7, we analyzed the performance of our model as a function of the number of labels in the source domain. We investigated how the number of labels in the source domain affected the performance of our model. In Figure 8, we plot the prediction scores obtained across multiple scripted cross-prioception domains, averaged across various source domains. The shaded region in this figure represents the amount of variance obtained when utilizing various source domains. Small regions of shading indicate that the scores are highly consistent across experiments. We observed a substantial difference when the target is “Phone in Bag” versus “Phone in Hand” or “Phone in Pocket”. Table 7 provides a more detailed version of this experiment. *Triple-DARE* attains superior prediction scores on the target domain using a small number of source labels, outdoing baseline methods in nearly all UDA tasks.

*(6) Intra-class compactness and inter-class separation*: To quantify the compactness and separation of learned feature embeddings, we employed the $\mathrm{Silhouettescore}\;\mathrm{Score}=\frac{b_{i}-a_{i}}{\max\left(b_{i},a_{i}\right)}$, where $b_{i}$ is the shortest average distance between a point and every other point in any cluster, whereas $a_{i}$ represents the average distance between *i* and all data points belonging to the same cluster. This score accounts for both compactness and separation. To compute the Silhouette scores for the learned feature embeddings, we assign each instance one of the binary context labels as a cluster label. Then, we calculate the mean score across labels. In most UDA tasks, our *Triple-DARE* method achieves higher compactness and separation scores, as shown in Figure 9. In addition, in the majority of instances, CORAL achieves higher scores than DAN in most cases. Additionally, the quality of the learned feature embeddings can be viewed visually in Figure 10, which depicts the same context instances represented by feature embeddings learned using DAN and *Triple-DARE*. The visualization is obtained by projecting feature embeddings into a two-dimensional space using the T-distributed Stochastic Neighbor Embedding (TSNE) [32].

*(7) Ablation Study*: We conducted an experimental ablation (shown in Figure 11) to rank the utility of *Triple-DARE* for a variety of UDA tasks. The best results were seen when using all its parts together. To further understand the relative impact of each component in this ablation investigation, we employed a non-pretrained HCR model. While the triplet loss and the domain loss are both useful, they do not provide as much insight as joint training.

## 6. Limitations and Future Work

The assumption that the same number of sensors are available in scripted datasets and in in-wild datasets is one limitation of our methodology. We plan to investigate the possibility of using algorithms for lab-to-field recognition that utilize just a small fraction of sensors that are comparable in both domains. Increasing the model’s resilience against missing sensors during inference is one way that our methodology might be improved. We hope that future studies in visual analytics will make use of our proposed strategy for representation learning for smartphone sensor data and the use of UDA for visualization.

## 7. Conclusions

The performance of machine learning HCR models on real-world datasets is hindered by diverse phone placements and smartphone models, as well as weak, noisy, or missing labels. The goal of lab-to-field methods is to improve the performance of HCR models by first training them on scripted HCR datasets and then modifying them so that they can be used for predicting context labels in comparable datasets that were collected in the wild. This is the first work we are aware of that uses lab-to-field techniques on HCR datasets collected from smartphones. This paper presents *Triple-DARE*, a UDA deep-learning model for HCR on smartphones, which is comprised of three components: (1) a domain alignment loss that utilizes MK-MMD (2) a classification loss and (3) a joint-fusion triplet loss particularly designed for multi-labeled datasets. *Triple-DARE* learns domain-invariant features common to both datasets, decreasing the influence of noisy in-the-wild data by concentrating on salient areas in sensor inputs, and achieving a high F1-score for multiple UDA tasks on both scripted and in-the-wild context datasets. With its domain alignment loss, *Triple-DARE* outperforms state-of-the-art baseline approaches when it comes to mapping the source and target feature embedding into a standard feature distribution. In addition, the triplet loss improves discrimination by increasing intra-class compactness and inter-class separation while utilizing enormous amounts of unlabeled data. *Triple-DARE* outperforms other state-of-the-art DA baselines, increasing the F1-score and classification accuracy by 4.6% and 1.89%, respectively, and outperforming models with no adaptations by 10.7% and 14.7%.

## Figures and Tables

**Figure 1 sensors-23-03081-f001:**
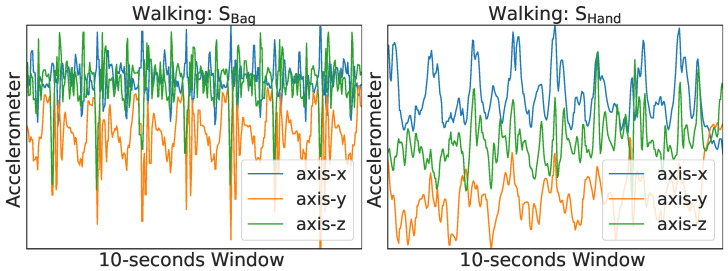
The effect of different phone placements on sensor data can be seen in triaxial accelerometer tracings for the same walking activity but with different phone prioceptions.

**Figure 2 sensors-23-03081-f002:**
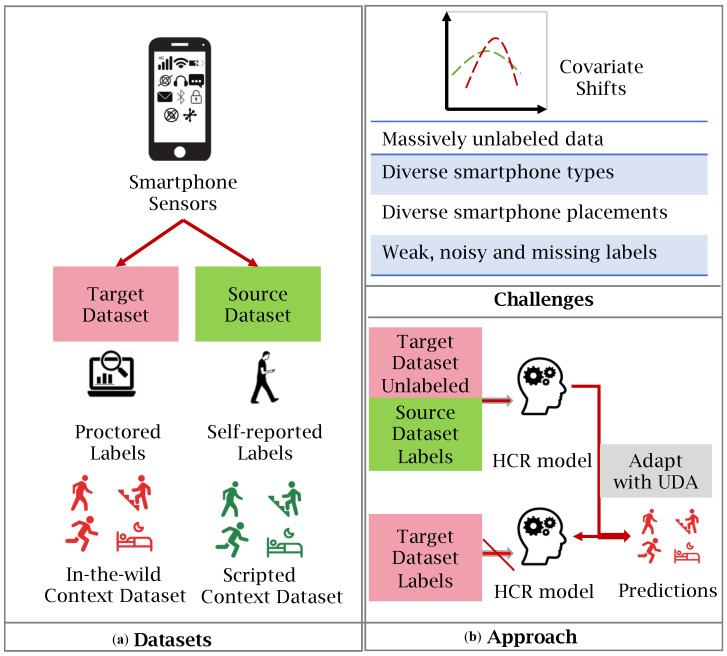
(**a**) The two kinds of smartphone context data used in this work. (**b**) Overview of *Triple-DARE*’s problem and approach.

**Figure 3 sensors-23-03081-f003:**
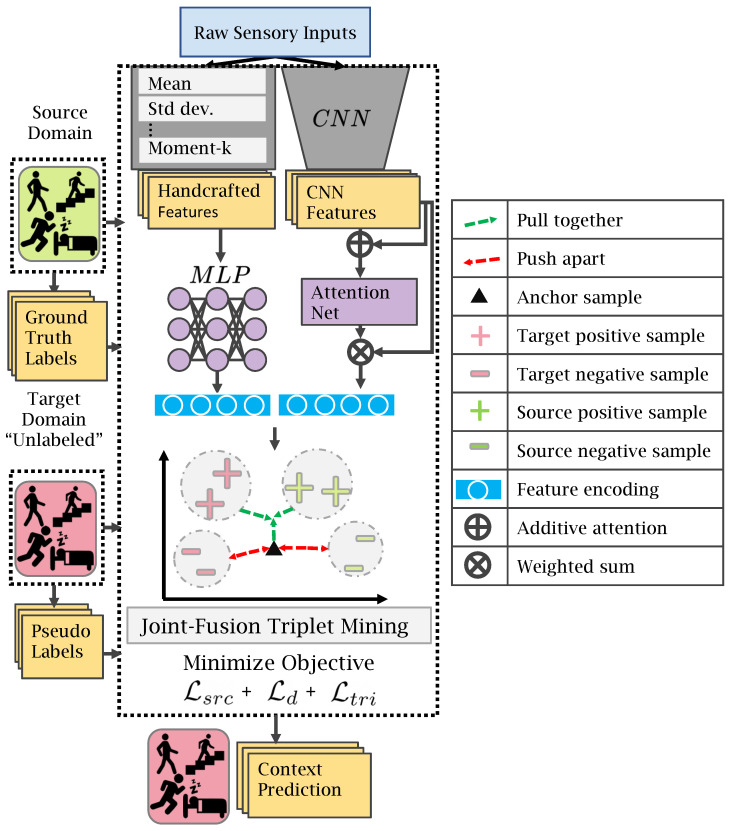
*Triple-DARE* framework.

**Figure 4 sensors-23-03081-f004:**
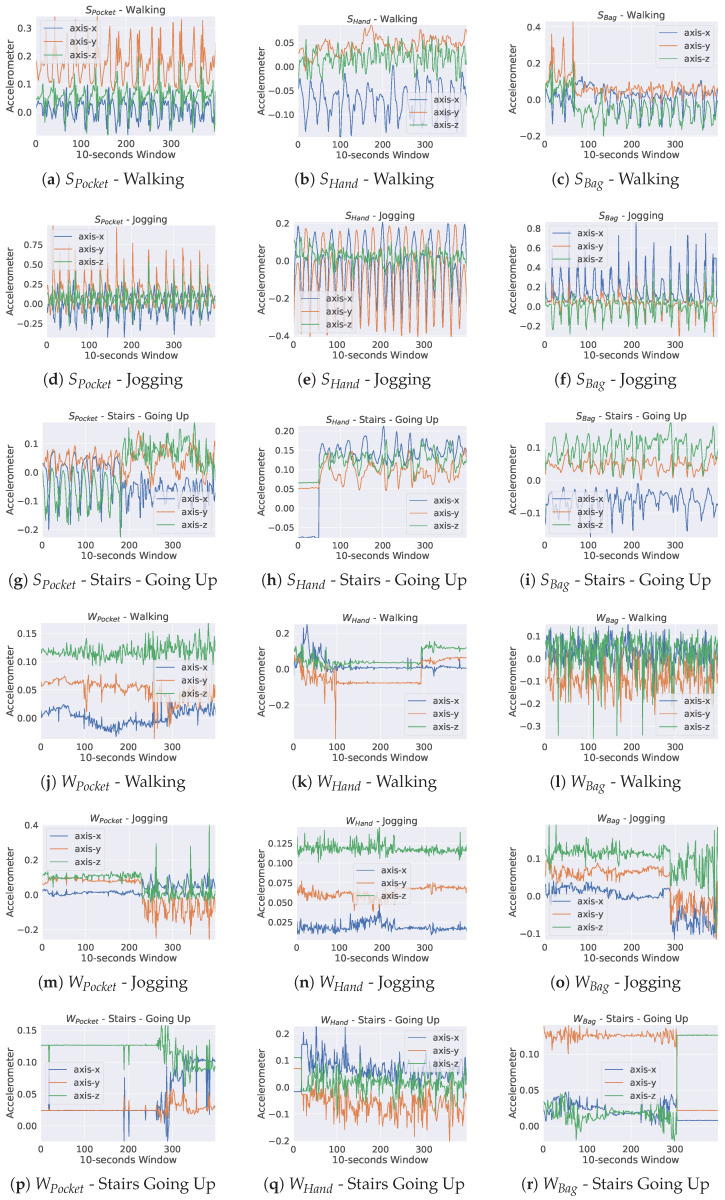
Raw accelerometer tracings sampled from Walking, Jogging, and Stairs Going up contexts within each dataset.

**Figure 5 sensors-23-03081-f005:**
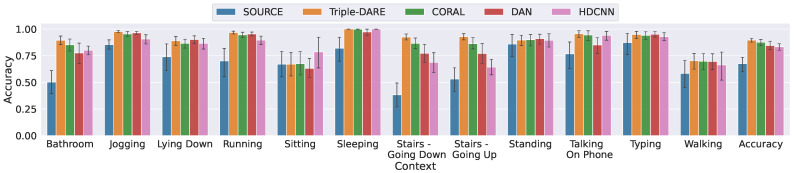
Target prediction scores for each label, averaged across various UDA task domains.

**Figure 6 sensors-23-03081-f006:**
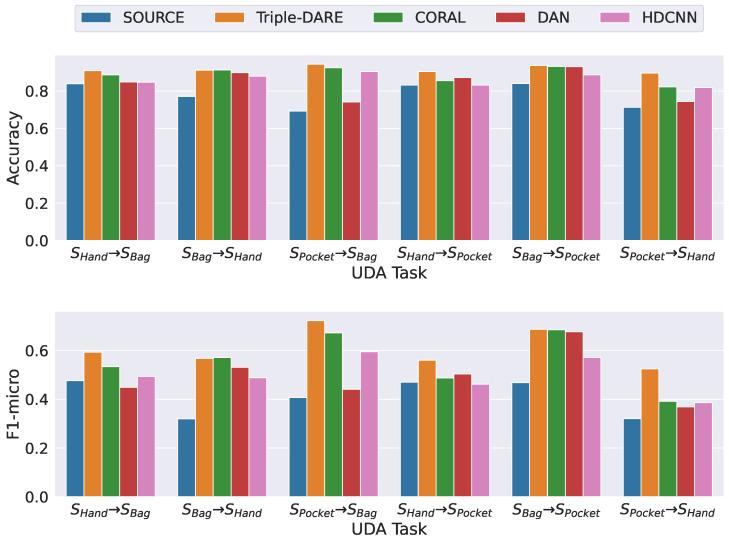
Scripted context data with cross-prioception UDA tasks.

**Figure 7 sensors-23-03081-f007:**
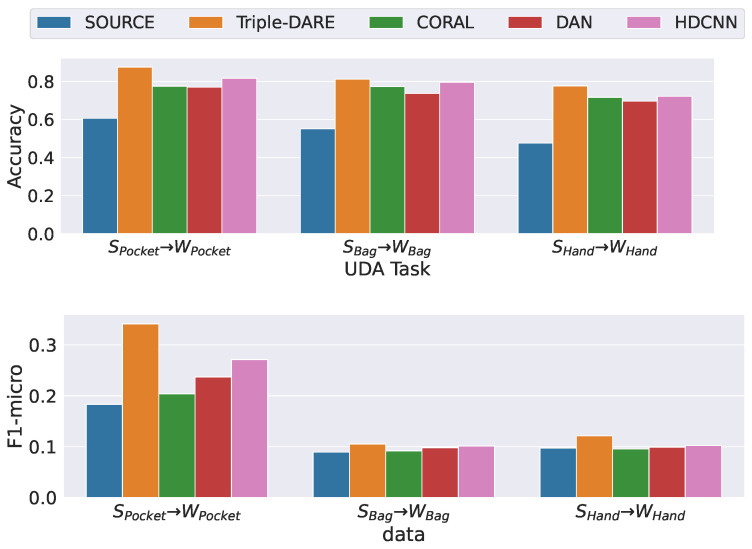
Scripted context to In-The Wild UDA tasks scores.

**Figure 8 sensors-23-03081-f008:**
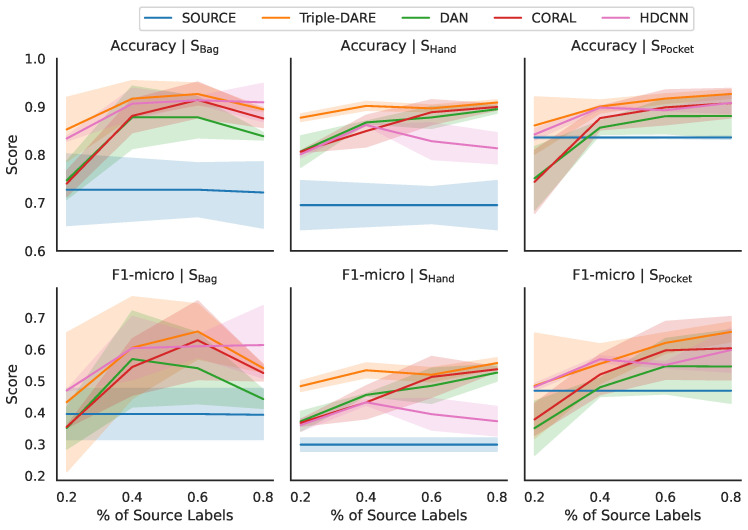
Scores for each source domain in scripted contexts with cross-prioception UDA tasks, averaged over each target, varying the number of labels from the source domain.

**Figure 9 sensors-23-03081-f009:**
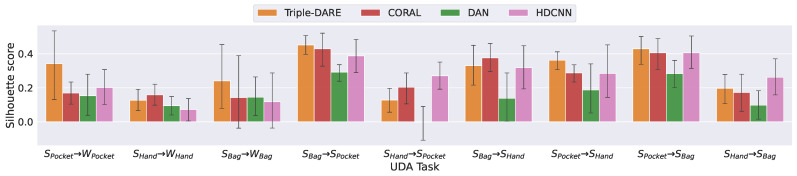
Compactness measure on feature embeddings.

**Figure 10 sensors-23-03081-f010:**
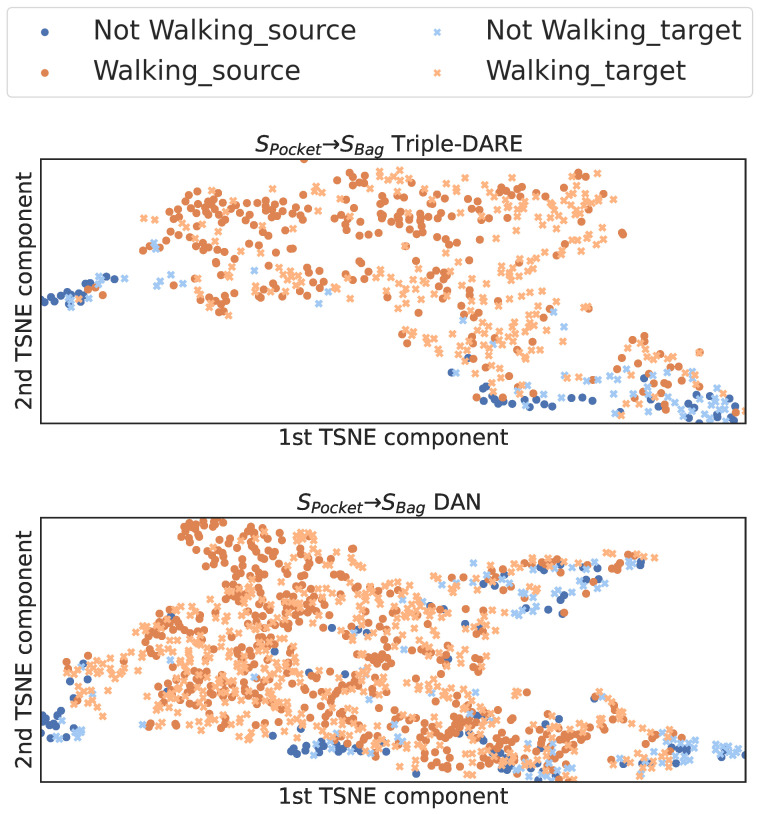
Visualization of the learned feature embeddings for TripleDARE (**top**) and DAN (**bottom**), using TSNE dimensional reduction.

**Figure 11 sensors-23-03081-f011:**
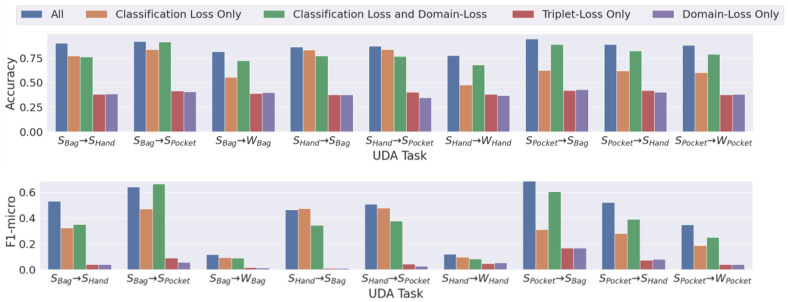
Ablation study, evaluating the contribution of *Triple-DARE*’s each component.

**Table 1 sensors-23-03081-t001:** Context-specific diagnostic tests for traumatic brain injury (TBI) and infectious diseases with their relevant human contexts.

Traumatic Brain Injury	
Diagnostic Test	Context
Inferior Reaction Time	< Interaction with Phone, in Hand, *, *>
Elevated Light Sensitivity	<*, in Hand, *, *>
Pupil Dilation	< Interaction w/ Phone, in Hand, Typing, *>
Hands Shaking	<*, in Hand, *, *>
Slurred Speech	<Speaking into Phone, *, *, *>
**Infectious Diseases**	
**Ailment Test**	**Test Context**
Elevated Frequency of Coughing	<Coughing, *, *, *>
Elevated Frequency of Sneezing	<Sneezing,*, *, *>
Rate of Heart at Rest	<Sitting, in Pocket, *, *>
Elevated Toilet use Frequency	<Using Toilet, *, *, *>
Variation in respiration	<Sleeping, on Table, *, *>
	<Exercising, *, *, *>
**Both TBI and Infectious Disease**	
**Ailment Test**	**Test Context**
Elevation In Activity Transition Period	<Lying down, in Pocket, *, *>
	<Sitting, in Pocket, *, *>
	<Standing, in Pocket, *, *>
Variation in Sleep Quality	<Sleeping, *, *, *>
Variation in Gait	<Walking, in Pocket/Hand, *, *>

“*” denoting a wildcard.

**Table 2 sensors-23-03081-t002:** Contexts for which data was obtained as part of our WASH Study Collected Contexts—split down into 25 binary labels.

Phone Placement	
Phone in Hand	Phone in Bag
Phone in Table—Facing Up	Phone in Table—Facing Down
Phone in Pocket	
**Long Activity**	
Standing	Sleeping
Walking	Sitting
Stairs—Going Up	Stairs—Going Down
Talking On Phone	Trembling ${}^{*}$
Jumping ${}^{*}$	Jogging
Typing	In Bathroom
Lying Down	Running
**Short Activity**	
Coughing ${}^{*}$	Sneezing ${}^{*}$
Sitting Down (transition) ${}^{*}$	Sitting Up (transition) ${}^{*}$
Standing up (transition) ${}^{*}$	Laying Down (transition) ${}^{*}$

*: Labels associated with contexts collected in the scripted study only.

**Table 3 sensors-23-03081-t003:** Related work summary.

Research Work	Method	Type of Data	Task	Lab-to-Field	Distribution Discrepancy Minimization
Natarajan et al. [9]	Importance-reweighting	Wearable electrocardiogram sensor data	Cocaine use detection	×	No
Alajaji et al. [1]	Positive Unlabeled Classifier	Smarthpone sensor data	Context recognition	×	No
Chang et al. [5]	Feature matching and confusion maximization	Wearable sensor data	UDA for activity recognition under sensor position variability		Global only
Long et al. [18]	MK-MMD	Images	UDA for cross-dataset image classification		Global only
Sun et al. [19]	Correlation Alignment	Images	UDA for cross-dataset image classification		Global only
Khan et al. [17]	KL Divergence	Smartphone and smartwatch sensor data	DA for cross-device activity recognition		Global only
Chen et al. [26]	Stratified Transfer Learning	Wearable sensor data	DA for cross sensor placement		Non-scalable intra-class separation
Sanabria et al. [27]	Variational Autoencoder	Binary event sensor data	DA for cross-user activity recognition		Global only
Wilson et al. [20]	Weak-supervision using target label distribution	Wearable sensor data	DA for cross-user activity recognition		Global only and utilized target labels

**Table 4 sensors-23-03081-t004:** A small selection of the handcrafted features applied to accelerometer, gyroscope, and magnetometer data that we use, taken from Refs. [3,29].

Feature	Formulation
Tri-axial sensors Features	
Arithmetic mean	$\bar{s}=\frac{1}{N}\sum_{i=1}^{N}s_{i}$
Standard deviation	$\sigma =\sqrt{\frac{1}{N}\sum_{i=1}^{N}{\left(s_{i}-\bar{s}\right)}^{2}}$
Frequency signal Skewness	E $\left[\frac{{\left(s-\bar{s}\right)}^{3}}{\sigma }\right]$
Frequency signal Kurtosis	$E\left[{\left(s-\bar{s}\right)}^{4}\right]/E{\left[{\left(s-\bar{s}\right)}^{2}\right]}^{2}$
Signal magnitude area	$\frac{1}{3}\sum_{i=1}^{3}\sum_{j=1}^{N}\left|s_{i,j}\right|$
Pearson Correlation coefficient	$C_{1,2}/\sqrt{C_{1,1}C_{2,2}},C=\mathrm{cov}\left(s_{1},s_{2}\right)$
Spectral energy of a frequency band [a, b]	$\frac{1}{a-b+1}\sum_{i=a}^{b}s_{i}^{2}$
s: signal vector Q: quartile, N: signal vector length, cov: covariance	
GPS Features	
Significant changes from the prior location state	
Estimated speed	
Variations in latitude and longitude	
Phone State Features	
Is smartphone screen unlocked?	Is smartphone charging?
Is ringer setting set to silent?	Is smartphone connected to WIFI?

**Table 5 sensors-23-03081-t005:** The proportions of contexts that have been given a positive label.

Contexts	Scripted % P	In-the-Wild % P
Bathroom	3.15%	2.17%
Jogging	2.04%	0.27%
Lying Down	1.10%	16.24%
Running	1.95%	0.37%
Sitting	11.99%	38.71%
Sleeping	2.19%	37.69%
Stairs—Going Down	2.52%	2.00%
Stairs—Going Up	0.89%	1.92%
Standing	1.71%	8.46%
Talking On Phone	1.41%	1.27%
Typing	3.65%	6.45%
Walking	64.00%	13.51%
**Phone Prioceptions**		
Phone In Hand	Phone In Pocket	Phone In Bag
**Datasets Notations**		
$S_{\mathrm{Prioception}}$	Scripted context dataset	
$W_{\mathrm{Prioception}}$	In-the-wild context dataset	
e.g., $S_{\mathrm{Pocket}}$ refers to scripted contexts, annotated with “Phone In Pocket”		

**Table 6 sensors-23-03081-t006:** Overall context prediction scores on the target domain.

Overall UDA Tasks	Accuracy	F1-micro
*Triple-DARE*	**0.879**	**0.366**
CORAL	0.806	0.302
DAN	0.673	0.294
HDCNN	0.816	0.3215
Source (no adaptation)	0.433	0.259
Lab-to-field UDA Tasks	Accuracy	F1-micro
*Triple-DARE*	**0.845**	**0.188**
CORAL	0.839	0.127
DAN	0.698	0.122
HDCNN	0.768	0.146
Source (no adaptation)	0.552	0.133

*The highest scores are highlighted in bold font. Row colors match the colors used for different baseline methods throughout this article.*

**Table 7 sensors-23-03081-t007:** F-1 scores—Using variable amounts of source labels to compare diverse UDA methods for a variety of tasks.

	Scripted Contexts with Cross-Prioception UDA Tasks								Lab-to-Field UDA Tasks			
Training %	Method	$S_{\mathrm{Bag}}$→$S_{\mathrm{Hand}}$	$S_{\mathrm{Bag}}$→$S_{\mathrm{Pocket}}$	$S_{\mathrm{Hand}}$→$S_{\mathrm{Bag}}$	$S_{\mathrm{Hand}}$→$S_{\mathrm{Pocket}}$	$S_{\mathrm{Pocket}}$→$S_{\mathrm{Bag}}$	$S_{\mathrm{Pocket}}$→$S_{\mathrm{Hand}}$	Avg	$S_{\mathrm{Bag}}$→$W_{\mathrm{Bag}}$	$S_{\mathrm{Hand}}$→$W_{\mathrm{Hand}}$	$S_{\mathrm{Pocket}}$→$W_{\mathrm{Pocket}}$	Avg
0.2	*Triple-DARE*	**0.500**	**0.651**	0.213	0.318	**0.652**	**0.467**	**0.467**	**0.101**	0.080	**0.326**	**0.169**
	CORAL	0.357	0.328	0.357	0.428	0.352	0.378	0.367	0.089	**0.087**	0.150	0.109
	DAN	0.341	0.436	0.285	0.265	0.418	0.403	0.358	0.079	0.077	0.165	0.107
	HDCNN	0.341	0.492	**0.472**	**0.470**	0.468	0.380	0.437	0.087	0.084	0.181	0.117
0.4	*Triple-DARE*	**0.557**	**0.617**	0.444	**0.492**	**0.767**	**0.511**	**0.565**	**0.118**	**0.143**	**0.359**	**0.207**
	CORAL	0.380	0.584	**0.455**	0.457	0.633	0.484	0.499	0.092	0.075	0.165	0.111
	DAN	0.452	0.509	0.418	0.451	0.721	0.459	0.502	0.101	0.093	0.244	0.146
	HDCNN	0.424	0.580	0.504	0.558	0.704	0.441	0.535	0.106	0.108	0.266	0.160
0.6	*Triple-DARE*	0.497	0.588	**0.570**	**0.653**	0.744	**0.542**	**0.599**	**0.111**	0.112	**0.341**	**0.188**
	CORAL	**0.577**	**0.688**	0.505	0.505	**0.754**	0.448	0.580	0.110	**0.123**	0.210	0.148
	DAN	0.540	0.634	0.428	0.459	0.653	0.429	0.524	0.100	0.084	0.209	0.127
	HDCNN	0.345	0.561	0.575	0.540	0.645	0.445	0.518	0.094	0.102	0.285	0.160
Average	*Triple-DARE*	**0.518**	**0.619**	0.409	0.488	**0.721**	**0.507**	**0.544**	**0.111**	**0.112**	**0.341**	**0.188**
	CORAL	0.438	0.533	0.439	0.463	0.580	0.437	0.482	0.097	0.093	0.173	0.122
	DAN	0.440	0.526	0.377	0.392	0.597	0.430	0.461	0.096	0.087	0.198	0.127
	HDCNN	0.370	0.544	**0.517**	**0.523**	0.606	0.422	0.497	0.096	0.098	0.244	0.146
-	No Adaptation	0.319	0.469	0.476	0.470	0.260	0.315	0.385	0.108	0.110	0.180	0.133

*The highest scores are highlighted in bold font. Row colors match the colors used for different baseline methods throughout this article.*

## Data Availability

The data presented in this study are available on request from the corresponding author. The data are not publicly available due to privacy agreements.

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
