# Peer review of "Domain Adaptation Methods for Lab-to-Field Human Context Recognition†"

_sensors, 2023, doi:10.3390/s23063081_

Round 1
Reviewer 1 Report
Authors have done a good job by presenting this article in well organised manner, however there are some concerns to be addressed.
Introduction - Including motivation behind this problem & research makes this section complete.
Background & Related Work - Please include the gaps identified in existing methods as well in Table 3.
Methodology - Mathematical formulations can be explained more clearly so that reader can understand easily without facing it difficulty.
Results - Add samples of dataset as a figure if possible and discuss the results statistically. Also include ablation studies if done for this research.
Conclusion - Add future directions & limitations of proposed work if any.
Highlight the novelty of the work proposed. Overall, this article is interesting to read and well presented.
Reviewer 2 Report
The authors researched Triplet-based Domain Adaptation for Context Recognition, with novel idea and better performance. The paper is well-written and rigorous experiments prove its performance. Graphs and data are also faithfully included, and the effect of the proposed method is well explained with ablation study.
Reviewer 3 Report
47% plagiarism detected.
for further proceeding the article first author have to include novality and remove plagiarism.
the paper is good but took plagiarism, i agreed to publish this article.All the reference is used in the manuscript should be in order form. In this manuscript mostly the reference was used randomly.
In addition, the conclusions and abstract must be self-contained.
All the used diagrams and table description is too long, it should be short, such as
"Figure 1: a) The nature of the two smartphone context data we use in this work. b) Ahigh-level overview for Triple-DARE s problem and approach" it should be in short form. Figure1: (A) Two SmartPhone Context Data while (B) Overview of Triple-DARE Problem and Approach. i have no more technical comments.

Reviewer 4 Report
The manuscript entitled “Domain adaptation methods for Lab-to-field human context recognition” is an extension of the conference paper entitled “Triplet-based domain adaptation (Triple-Dare) for Lab-to-field human context recognition, IEEE”. The manuscript proposes a Triple-Dare, an unsupervised domain adaptation deep learning model for human context recognition. Authors claim that the model outperforms other models reported in the literature. The document is quite interesting, and relevant. The document is properly structured providing the necessary background, and definitions to understand the problem. Along the document, diverse figures and tables help to understand the results and other relevant information. Most of the bibliographic references can be considered recent. I have no problem or further comments on this contribution. My recommendation is to accept.
Round 2
Reviewer 3 Report
Approved